# Stochastic No-regret Learning for General Games with Variance Reduction

**Yichi Zhou**
Microsoft Research, AI4Science, Asia
Beijing, China
yiczho@microsoft.com

**Fang Kong & Shuai Li**
Shanghai Jiao Tong University
Shanghai, China
{fangkong,shuaili8}@sjtu.edu.cn

## ABSTRACT

We show that a stochastic version of optimistic mirror descent (OMD), a variant of mirror descent with recency bias, converges fast in general games. More specifically, with our algorithm, the individual regret of each player vanishes at a speed of $O(1/T^{3/4})$ and the sum of all players' regret vanishes at a speed of $O(1/T)$, which is an improvement upon the $O(1/\sqrt{T})$ convergence rate of prior stochastic algorithms, where $T$ is the number of interaction rounds.

Due to the advantage of stochastic methods in the computational cost, we significantly improve the time complexity over the deterministic algorithms to approximate coarse correlated equilibrium. To achieve lower time complexity, we equip the stochastic version of OMD in (AM21) with a novel low-variance Monte-Carlo estimator. Our algorithm extends previous works (AM21; CJST19) from two-player zero-sum games to general games.

## 1 INTRODUCTION

How does a player in a game interact with others, and selfishly maximize its own utilities? This is one central problem in online learning and game theory and has intimate connections to economics, auction design, and machine learning. The study of this problem was pioneered by (Bro49; Rob51). Robinson (Rob51) shows that fictitious play asymptotically converges to Nash equilibrium in two-player zero-sum games. But its convergence rate is exponentially slow and it may not even converge in non-zero-sum games (Sha64).

Another natural choice for each player is to use no-regret learning algorithms. With some well-known families of no-regret learning algorithms, e.g., mirror descent (NY83) and follow-the-regularized-leader (KV05), the average regret of each player vanishes at a speed of $O(1/\sqrt{T})$ where $T$ is the number of interaction rounds. This regret bound implies an $O(1/\sqrt{T})$ convergence rate to the coarse correlated equilibrium in general games (or Nash equilibrium in two-player zero-sum games). And it is noteworthy that Chen and Peng (CP20) show that these algorithms' convergence rate is $\Omega(1/\sqrt{T})$.

Players can do even better with some special no-regret algorithms tailored for games. The most representative one is known as optimistic mirror descent (OMD) which is a variant of mirror descent with recency bias. Syrgkanis et al. (SALS15) and Rakhlin et al. (RS13) show that OMD approaches optimal social welfare (or equivalently, minimizes the sum of all players' regrets) at a speed of $O(1/T)$ and minimizes each player's individual regret at a speed of $O(1/T^{3/4})$. Several works (CP20; HAM21; DFG21) improve the results in (SALS15) under different settings or assumptions. Remarkably, Daskalakis et al. (DFG21) improve the convergence rate of players' individual regret of OMD to $O(\text{poly} \log T/T)$ in general games.

However, the computational cost of players to use OMD, as well as other deterministic no-regret algorithms, could be not manageable. Since each player needs to compute the exact loss vector to update its strategy in OMD. And the time complexity of computing this exact loss vector is, in the worst case, exponential in the number of players in the game. One standard method to accelerate the computation is to estimate the loss vector with Monte-Carlo methods. But a Monte-Carlo estimator with an uncontrolled variance will immediately make the convergence rate degenerate to $O(1/\sqrt{T})$.

To alleviate the effect of Monte-Carlo estimator's variance, Carmon et al. (CJST19) and Alacaoglu et al. (AM21) propose variance reduced stochastic no-regret algorithms with a convergence rate of $O(1/T)$ for two-player zero-sum games. As a result, they improve the time complexity of computing $\epsilon$-Nash Equilibrium in two-player zero-sum games from $O(\text{Cost}/\epsilon)$ of deterministic algorithms to $O(\text{Cost} + \sqrt{\text{Cost}}/\epsilon)$ (some lower order terms are omitted) where Cost is the time complexity of computing the loss vector.

While Carmon et al. (CJST19) and Alacaoglu et al. (AM21) make a huge step towards developing efficient stochastic algorithms for games, their algorithms are tailored for the simplest two-player zero-sum games and could not cover more practical settings, such as auctions, which may involve multiple players and not be zero-sum. One crucial factor of the algorithms in Carmon et al. (CJST19) and Alacaoglu et al. (AM21) is a stochastic loss estimator with small variance. However, the time complexity of calculating this estimator is $O(A^{N-1})$ where $N$ is the number of players, which is exponentially large in general games. The high complexity of the estimator becomes a major obstacle to developing efficient stochastic algorithms for general games.

**Contributions.** We consider general normal-form games with an arbitrary number of players. Compared to the two-player zero-sum case, this is more challenging and practically significant. We show that in general games, a stochastic version of OMD converges to the optimal social welfare (or equivalently, minimizes the sum of all players' regrets) at a rate of $\tilde{O}(1/T)$ and minimizes the individual regret at a speed of $O(1/T^{3/4})$ in contrast to the $O(1/\sqrt{T})$ convergence rate of existing stochastic algorithms. Due to the advantage of stochastic methods in the computational cost, this significantly improves the time complexity to approximate coarse correlated equilibrium in general games. Please see Table 1 for the comparison of the time complexity of our algorithm against prior works. Specifically, our result improves previous works for weak $\epsilon$-CCE when $\text{Cost} \geq NA$ and for strong $\epsilon$-CCE when $\text{Cost} \geq NA^2/\epsilon$.

To achieve the above regret bounds, we make two main technical contributions. Firstly, we extend the theoretical framework of analyzing regret bounds of stochastic OMD in (AM21) from two-player zero-sum games to general games. Secondly, we propose a novel low-variance Monte-Carlo estimator for general games. The computational complexity of this estimator is exponentially faster than Carmon et al. (CJST19) and Alacaoglu et al. (AM21) while the variance is only slightly larger. The stochastic OMD algorithm equipped with our novel estimator achieves the above results.

The rest of the paper is organized as follows: In Section 2, we discuss prior works related to this problem. In Section 3, we provide necessary preliminaries for games, coarse correlated equilibrium and optimistic mirror descent. In Section 4, we introduce our algorithm and present a general regret upper bound in Theorem 1. In Section 5, we introduce our low-variance Monte-Carlo estimator and analyze its variance in Lemma 3. In Section 6, by combining the results in Theorem 1 and Lemma 3, we present our final regret bounds in Theorem 2 and 3, as well as the time complexity to approximate coarse correlated equilibrium in Corollary 1 and 2.

## 2 RELATED WORK

**Comparisons to existing algorithms.** Table 1 compares the time complexity of our algorithm to compute $\epsilon$-coarse correlated equilibrium for general games (and $\epsilon$-Nash equilibrium for two-player zero-sum games) against prior no-regret algorithms. The time complexity is determined by two terms: the convergence rate (or the regret) and the computational cost in each round. Deterministic algorithms (PSS21; DFG21; SALS15) converge fast, but with a relatively higher per round time complexity since they have to compute the loss in each round. Stochastic algorithms exploit the Monte-Carlo approach to accelerate the computation of loss. However, the variance of the estimated loss may slow down the convergence rate. To alleviate the effect of the variance, Carmon et al. (CJST19) and Alacaoglu et al. (AM21) develop variance reduced stochastic no-regret learning algorithms. Their algorithms significantly accelerate the computation of $\epsilon$-Nash equilibrium in two-player zero-sum games.

**Variance reduction.** Variance reduction is one of the most useful techniques to accelerate stochastic algorithms (see (GSBR20) for a comprehensive survey). Typically, when optimizing the finite sum problem $\min_x F(x) = \sum_{i=1}^{N} F_i(x)$, instead of estimating the gradient by $\nabla F_i(x)$ like Stochastic Gradient Descent (SGD), the variance reduction method proposes to use $\tilde{\nabla} F(x) = \nabla F(w^k) +$

|  | $\epsilon$-Nash equilibrium
Two-player zero-sum | Weak $\epsilon$-CCE
General games | Strong $\epsilon$-CCE
General games |
|---|---|---|---|
| (D)(SALS15) | $\tilde{O}(\text{Cost}/\epsilon)$ | $\tilde{O}(N^3\text{Cost}/\epsilon)$ | $\tilde{O}(N^{3/2}\text{Cost}/\epsilon^{4/3})$ |
| (D)(DFG21) | $\tilde{O}(\text{Cost}/\epsilon)$ | $\tilde{O}(N^3\text{Cost}/\epsilon)$ | $\tilde{O}(N^2\text{Cost}/\epsilon)$ |
| (D)(PSS21) | $\tilde{O}(\text{Cost}/\epsilon)$ | $\tilde{O}(N^3\text{Cost}/\epsilon)$ | $\tilde{O}(N^2\text{Cost}/\epsilon)$ |
| (S)(AM21) | $\tilde{O}(\text{Cost} + \sqrt{A\text{Cost}}/\epsilon)$ | - | - |
| (S)(CJST19) | $\tilde{O}(\text{Cost} + \sqrt{A\text{Cost}}/\epsilon)$ | - | - |
| (S) Ours | $\tilde{O}(\text{Cost} + \sqrt{A\text{Cost}}/\epsilon)$ | $\tilde{O}(\text{Cost} + N^{\frac{7}{2}}\sqrt{A\text{Cost}}/\epsilon)$ | $O(\text{Cost} + \text{Cost}^{\frac{2}{3}}\frac{N^{\frac{7}{3}}A^{\frac{2}{3}}}{\epsilon^{4/3}})$ |

Table 1: Comparisons of time complexity to compute $\epsilon$-CCE for general games and $\epsilon$-Nash equilibrium for two-player zero-sum games. Cost denotes the time complexity of computing the loss vector. $N$ is the number of players and $A$ is the number of actions for each player. (S) means the algorithm is stochastic and (D) means the algorithm is deterministic. For stochastic algorithms, the time complexity is the expected running time to achieve an expected approximation error, which directly follows (AM21). $\tilde{O}$ hides factors which are polynomial in terms of $\log(1/\epsilon)$ and $\log(A)$.

$\nabla F_i(x) - \nabla F_i(w^k)$, where $w^k$ is called the "snapshot". In many cases, e.g., when $F_i$'s are convex, sampling $i$ from a uniform distribution is sufficient to reduce the variance. When dealing with games, it turns out that one has to be meticulous to design a low-variance sampling distribution, even when the game is two-player zero-sum (CJST19; AM21). And it remains unclear how to design such a low-variance distribution for general games. One of our main contributions is a low-variance Monte-Carlo estimator for general games, which ensures fast convergence in general games.

**Stochastic no-regret learning for large-scale sequential decision-making.** Counterfactual regret minimization (CFR) (ZJBP07) generalizes no regret learning to games with sequential decision-making, namely extensive-form games. A long list of works study the variants of CFR (LWZB09; BS19a; ZRL+19; MSB+17; TBJB15). And these works lead to the superhuman AI, Libratus (BS18) and Pluribus (BS19b), for two-player and six-player heads-up no-limit Texas Hold'em (HUNL). Since the state space of HUNL is super huge, both Libratus and Pluribus adopt the Monte-Carlo CFR with external sampling (LWZB09). And the experimental results suggest that the variance of Monte-Carlo sampling significantly slows down the convergence rate. Several works empirically reduce the variance of Monte-Carlo sampling (SBL+19; BS19b). However, these algorithms do not enjoy any theoretical guarantees on the convergence rate. We hope our algorithms could provide insights into the development of low-variance stochastic algorithms for large-scale games.

## 3 PRELIMINARY

**Notation.** For any vector $v \in \mathbb{R}^d$, denote $v(j)$ as its $j$th coordinate, $\|v\|_1 = \sum_{j=1}^{d} |v(j)|$ as its $\ell_1$-norm, $\|v\|_2 = \sqrt{\sum_{j=1}^{d} v^2(j)}$ as its $\ell_2$-norm, and $\|v\|_\infty = \max_{j=1,\cdots,d} |v(j)|$ as its $\ell_\infty$ norm. For a general norm $\|\cdot\|$, let $\|\cdot\|_*$ represent its dual norm. Denote $\langle v, w \rangle = \sum_{i=1}^{d} v(i)w(i)$ as the standard inner product of two vectors $v, w \in \mathbb{R}^d$. For a positive integer $n$, let $[n] = \{1, \cdots, n\}$. For a discrete set $S$, let $\Delta(S)$ be the set of distributions over $S$.

**Basics of game.** We consider general games with $N$ players. The action space of each player $i \in [N]$ is $\mathcal{A}_i$. Denote $A = \max_i |\mathcal{A}_i|$ as the cardinality of the largest action space. The joint action space of all players is $\mathcal{A} = \mathcal{A}_1 \times \mathcal{A}_2 \times \cdots \times \mathcal{A}_N$. For simplicity, let $\mathcal{A}_{-i} = \mathcal{A}_1 \times \cdots \times \mathcal{A}_{i-1} \times \mathcal{A}_{i+1} \times \cdots \times \mathcal{A}_N$ be the joint action space of all players except for player $i \in [N]$. The loss of players can be specified by the functions $F_1, F_2, \cdots, F_N : \mathcal{A} \to [0, 1]$, which map the joint action space to a real value. Specifically, if each player $j$ selects action $a_j \in \mathcal{A}_j$, then $F_i(a)$ is the loss of player $i$ where $a := (a_1, a_2, \ldots, a_N)$.

A mixed strategy $\sigma_i$ is a probability distribution over $\mathcal{A}_i$. We say a player $p_i$ plays according to $\sigma_i$ if it selects action $a_i \in \mathcal{A}_i$ with probability $\sigma_i(a_i)$. For any strategy profile $\sigma := (\sigma_i)_{i \in [N]}$, let $\sigma_{-i} := (\sigma_1, \ldots, \sigma_{i-1}, \sigma_{i+1}, \ldots, \sigma_N)$ denote the strategy profile $\sigma$ after removing $\sigma_i$. And for convenience, let $(\sigma_i', \sigma_{-i})$ denote the strategy profile $\sigma$ after replacing $\sigma_i$ with $\sigma_i'$. Similarly, given an action profile $a := (a_i)_{i \in [N]}$, let $a_{-i} := (a_1, \ldots, a_{i-1}, a_{i+1}, \ldots, a_N)$ denote the action profile after removing $a_i$ and $(a_i', a_{-i})$ be the action profile $a$ after replacing $a_i$ with $a_i'$.

With a little abuse of notation, for a strategy profile $\sigma$, let $F_i(\sigma) := \mathbb{E}_{a \sim \sigma}\left[F_i(a)\right]$ be the expected loss of player $i$ if each player $j \in [N]$ plays according to $\sigma_j$. For convenience, define the vector $F_i(a_{-i}) \in [0,1]^{|\mathcal{A}_i|}$ with $[F_i(a_{-i})](a_i') = F_i((a_i', a_{-i}))$ representing the loss of player $i$ when each player $j \neq i$ selects $a_j$ and $i$ selects $a_i' \in \mathcal{A}_i$. Similarly, define the vector $F_i(\sigma_{-i}) \in [0,1]^{|\mathcal{A}_i|}$ with $[F_i(\sigma_{-i})](a_i') = \mathbb{E}_{a_{-i} \sim \sigma_{-i}}\left[F_i((a_i', a_{-i}))\right]$ representing the expected loss of player $i$ if each player $j$ plays according to $\sigma_j$ and $i$ selects action $a_i'$. Further, denote Cost as the time complexity of computing the vector $F_i(\sigma_{-i})$.

**No-regret learning.**  In online learning, $N$ players play the game for $T$ rounds. In round $k \in [T]$, player $i$ plays according to the strategy $\sigma_i^k$ and suffers the expected loss $\langle F_i(\sigma_{-i}^k), \sigma_i^k \rangle$. Each player aims to minimize its cumulative loss, which is equivalent to minimizing its regret $\max_{\sigma_i} R_i(\sigma_i) := \max_{\sigma_i} \sum_{k=1}^{T} \langle F_i(\sigma_{-i}^k), \sigma_i^k - \sigma_i \rangle$ where $R_i(\sigma_i)$ is the cumulative loss difference between the adopted strategies and the fixed strategy $\sigma_i$.

Given the strategy at round $1, ..., k$, the optimistic mirror descent (OMD) (RS13; SALS15) method calculates the strategy in round $k + 1$ as:

$$\sigma_i^{k+1} = \underset{\sigma_i \in \Delta(\mathcal{A}_i)}{\arg\min} D(\sigma_i, \sigma_i'^{k+1}), \text{ where } \nabla h(\sigma_i'^{k+1}) = \nabla h(\sigma_i^k) - \tau(2F_i(\sigma_{-i}^k) - F_i(\sigma_{-i}^{k-1}))). \quad (1)$$

where $\tau$ is the step size and $D(x, y)$ is the Bregman divergence induced by some mirror map $h(\cdot)$ [1]. Without loss of generality, we mainly consider two common mirror maps in this paper: negative entropy $h_1(x) = \sum_{a=1}^{d} x(a) \log x(a)$ and squared $\ell_2$-norm $h_2(x) = \sum_{a=1}^{d} x^2(a)$. Their corresponding Bregman divergences are $D_1(x, y) = \sum_{a=1}^{d} x(a) \log(x(a)/y(a))$ and $D_2(x, y) = \sum_{a=1}^{d}(x(a) - y(a))^2$, respectively. Our analyses apply to other mirror maps as well.

**Coarse correlated equilibrium (CCE).**  A correlated strategy $\zeta$ is a distribution over the joint action space $\mathcal{A}$. We call $\zeta$ a coarse correlated equilibrium (CCE) if no player can benefit from unilaterally deviating from $\zeta$ given that others take actions according to $\zeta$. There are two versions of approximate CCE: one corresponds to the social welfare and is referred to as weak $\epsilon$-CCE; the other corresponds to the individual loss and is referred to as strong $\epsilon$-CCE. Intuitively, weak $\epsilon$-CCE states that the averaged difference among all players $i \in [N]$ between the expected loss $\mathbb{E}_{a \sim \zeta}\left[F_i(a)\right]$ by following $\zeta$ and the least expected loss $\min_{a_i^*} \mathbb{E}_{a \sim \zeta}\left[F_i((a_i^*, a_{-i}))\right]$ by deviating from $\zeta$ is no more than $\epsilon$. And strong $\epsilon$-CCE requires that the maximum difference among all players $i \in [N]$ between these two types of expected loss to be smaller than $\epsilon$. The formal definitions are as follows.

**Definition 1.** *A correlated strategy $\zeta$ is a weak $\epsilon$-CCE (corresponding to social welfare) if*

$$\sum_{i=1}^{N} \left( \mathbb{E}_{a \sim \zeta}\left[F_i(a)\right] - \min_{a_i^*} \mathbb{E}_{a \sim \zeta}\left[F_i((a_i^*, a_{-i}))\right] \right) \leq \epsilon$$

*and it is a strong $\epsilon$-CCE (corresponding to individual loss) if*

$$\max_{i \in [N]} \left( \mathbb{E}_{a \sim \zeta}\left[F_i(a)\right] - \min_{a_i^*} \mathbb{E}_{a \sim \zeta}\left[F_i((a_i^*, a_{-i}))\right] \right) \leq \epsilon.$$

Here for convenience, we simply multiply the average difference by $N$ as the sum over all players in the definition of Weak $\epsilon$-CCE. Thus a strong $\epsilon$-CCE is a weak $N\epsilon$-CCE. Strong-CCE ensures that each player will not suffer a large loss, and weak-CCE guarantees the convergence to the global optimal strategy within a large class of smooth games (SALS15). Since

---

[1]$h : \mathbb{R}^d \to \mathbb{R}$ is called a mirror map if $h$ is strongly convex with respect to some norm and $\nabla h(\mathbb{R}^d) = \mathbb{R}^d$.

this work mainly focuses on the stochastic method, we study the expected performance of the generated strategy. Specifically, a randomly generated correlated strategy $\zeta$ is called a weak $\epsilon$-CCE if $\sum_{i=1}^{N} \left( \mathbb{E}_{\zeta, a \sim \zeta} \left[ F_i(a) \right] - \min_{a_i^*} \mathbb{E}_{\zeta, a \sim \zeta} \left[ F_i((a_i^*, a_{-i})) \right] \right) \leq \epsilon$ and a strong $\epsilon$-CCE if $\max_{i \in [N]} \left( \mathbb{E}_{\zeta, a \sim \zeta} \left[ F_i(a) \right] - \min_{a_i^*} \mathbb{E}_{\zeta, a \sim \zeta} \left[ F_i((a_i^*, a_{-i})) \right] \right) \leq \epsilon$, where the expectation is additionally taken over the randomness of the generated strategy $\zeta$.

**Connection between no-regret learning and CCE.** The following Lemma 1 shows a well-known connection between no-regret learning and CCE.

**Lemma 1.** *Let $\sigma^1, \cdots, \sigma^T$ denote an arbitrary collection of strategy profiles and $\zeta(a) := \frac{1}{T} \sum_{k=1}^{T} \prod_{i=1}^{N} \sigma_i^k(a_i)$. It holds that*

$$\mathbb{E}_{a \sim \zeta} \left[ F_i(a) \right] - \min_{a_i^*} \mathbb{E}_{a \sim \zeta} \left[ F_i((a_i^*, a_{-i})) \right] \leq \max_{\sigma_i} R_i(\sigma_i)/T, \forall i \in [N]. \tag{2}$$

Lemma 1 shows that the running time to approximate CCE depends on both the regret and the running time in each round. Deterministic algorithms, such as OMD, though enjoy excellent regret bounds, take a long running time for computation in each round. In the next sections, we will show how to use the Monte-Carlo method to accelerate the calculations without significantly hurting the regret bound.

## 4 OPTIMISTIC MIRROR DESCENT WITH MONTE-CARLO ESTIMATION

In this section, we present our algorithm. We first introduce some necessary notations. Let $F_{i,a_{-i}}(\sigma_{-i}) = F_i(a_{-i})\sigma_{-i}(a_{-i})/q(a_{-i})$ where $a_{-i} \in \mathcal{A}_{-i}$ denotes a random variable drawn from a sampling distribution $q_{-i}$.

The algorithm is presented in Algorithm 1, which is a stochastic version of OMD shown in equation 1 with variance reduction. There are two crucial modifications over the vanilla OMD to accelerate calculation: one is on the estimated loss and the other is on the update starting points.

---

**Algorithm 1** Optimistic mirror descent with variance reduction

1: **Input:** hyper-parameters $p, \tau$ and $\alpha$; mirror map $h$
2: **Initialize:** $w_i^1(a) = 1/A$ and $\sigma_i^1(a) = 1/A$
3: **for** $k = 1, 2, \cdots, T$ **do**
4:     Sample $u$ from uniform distribution over $[0, 1]$.
5:     **for** $i = 1, \cdots, N$ **do**
6:         Compute $\hat{\sigma}_i^k$ such that $\nabla h(\hat{\sigma}_i^k) = \alpha \nabla h(\sigma_i^k) + (1 - \alpha)\nabla h(w_i^k)$.
7:         Sample $a_{-i}^k \sim \text{LVE}(i, \sigma^k, w^{k-1})$. {See section 5 for the definition of LVE.}
8:         Compute $\tilde{\sigma}_i^{k+1}$ such that

$$\nabla h(\tilde{\sigma}_i^{k+1}) = \nabla h(\hat{\sigma}_i^k) - \tau(F_i(w_{-i}^k) + F_{i,a_{-i}^k}(\sigma_{-i}^k) - F_{i,a_{-i}^k}(w_{-i}^{k-1})).$$

9:         Compute $\sigma_i^{k+1} = \arg\min_{\sigma_i \in \Delta(\mathcal{A}_i)} D(\sigma_i, \tilde{\sigma}_i^{k+1})$.
10:        Update $w_i^{k+1} = \sigma_i^{k+1}$ if $u < p$ and $w_i^{k+1} = w_i^k$ if $u \geq p$.
11:     **end for**
12: **end for**

---

**Estimated loss.** As can be seen in Line 8, Algorithm 1 updates the strategy using the estimated loss $F_i(w_{-i}^k) + F_{i,a_{-i}^k}(\sigma_{-i}^k) - F_{i,a_{-i}^k}(w_{-i}^{k-1})$ instead of $2F_i(\sigma_{-i}^k) - F_i(\sigma_{-i}^{k-1})$ in the vanilla OMD equation 1. By constructing appropriate distribution $q_{-i}$ to sample $a_{-i}^k$ (Line 7), it can be shown that the adopted estimated loss $F_{i,a_{-i}^k}(\sigma_{-i}^k) - F_{i,a_{-i}^k}(w_{-i}^{k-1})$ where $F_{i,a_{-i}}(\sigma_{-i}) = F_i(a_{-i})\sigma_{-i}(a_{-i})/q(a_{-i})$ and $a_{-i} \sim q_{-i}$ in the algorithm is an unbiased estimator of $F_i(\sigma_{-i}^k) - F_i(w_{-i}^{k-1})$ in the vanilla OMD. The objective to introduce such estimated loss is to reduce the computation complexity. Since $w^{k+1}$ is only updated with probability $p$ at each round (Line 10), the running time to compute $F_i(w_{-i}^k)$ over $T$ rounds is just $O(pT\text{Cost})$. It is also worth noting that such an estimator may bring high variance. So how to construct $q_{-i}$ to accelerate convergence while ensuring low variance is the key innovation of our algorithm. We postpone the descriptions and discussions on this estimator to Section 5.

**Different update starting points.** As in Line 8, Algorithm 1 updates $\sigma_i^{k+1}$ from the starting point $\hat{\sigma}_i^k$ instead of $\sigma_i^k$ in OMD equation 1. This modification was first introduced by (AM21) for two-player zero-sum games. We prove its effect in general games is to admit a faster learning rate in the case of maximizing social welfare. The detailed proof for this part can be seen in the full analysis.

It is remarkable that under special settings of hyper-parameters, our Algorithm 1 degenerates to standard OMD (SALS15) and stochastic OMD for two-player zero-sum games (AM21). Specifically, when setting $p = 1, \alpha = 1$ and using the exact value of $F_i(\sigma_{-i}^k) - F_i(w_{-i}^{k-1})$ in Line 8, Algorithm 1 is equivalent to standard OMD (SALS15). And when the underlying game is two-player zero-sum, our constructed unbiased estimator for $F_i(\sigma_{-i}^k) - F_i(w_{-i}^{k-1})$ is equivalent to that in (AM21) and thus Algorithm 1 can also recover their stochastic OMD.

In the following, we show how to compute $\sigma^{k+1}$ by taking two common examples of the mirror map function: negative entropy $h_1(x) = \sum_{a=1}^d x(a) \log x(a)$ and square of $\ell_2$-norm $h_2(x) = \sum_{a=1}^d x^2(a)$. When the mirror map is $h_1$, it is easy to verify that $\nabla_{x(a)} h_1(x) = 1 + \log x(a)$. Then according to Line 6, $\hat{\sigma}_i^k(a) = (\sigma_i^k(a))^\alpha (w_i^k(a))^{1-\alpha}$. So we can update $\sigma_i^{k+1}$ in $O(A)$ time as

$$\sigma_i^{k+1}(a) \propto (\sigma_i^k(a))^\alpha (w_i^k(a))^{1-\alpha} \exp\left(-\tau[F_i(w_{-i}^k) + F_{i,a_{-i}^k}(\sigma_{-i}^k) - F_{i,a_{-i}^k}(w_{-i}^{k-1})](a)\right).$$

Similarly, when the mirror map is $h_2$, we can directly compute $\hat{\sigma}_i^k(a) = \alpha \sigma_i^k(a) + (1-\alpha) w_i^k(a)$. The computation of $\tilde{\sigma}_i^{k+1}$ is easy according to the gradient of the $\ell_2$-norm and $\sigma_i^{k+1}$ can then be obtained by standard procedure of projecting $\tilde{\sigma}_i^{k+1}$ to the simplex.

## 4.1 THEORETICAL ANALYSIS

We first provide a general regret upper bound for Algorithm 1 in the following Theorem 1. The final results (Theorem 2 and Theorem 3) will be postponed in later sections by combining this general upper bound and the variance upper bound for the unbiased estimator in Lemma 3 as well as some delicate derivations.

**Theorem 1.** *If $D(x, y) \geq \gamma \|x - y\|^2$ for all $x, y$ and some norm $\|\cdot\|$ with $\gamma > 0$ being a constant, then the expected regret of Algorithm 1 is upper-bounded by*

$$\tau \max_{\sigma_i} \mathbb{E}\left[R_i(\sigma_i)\right] \leq U_i - (1-\alpha)\mathbb{E}\left[\sum_{k=1}^T D\left(\sigma_i^k, w_i^{k-1}\right)\right]$$

$$+ \tau^2\left(1 + \frac{1}{\alpha\gamma}\right)\mathbb{E}\left[\sum_{k=1}^T \|F_{i,a_{-i}^k}(\sigma_{-i}^k) - F_{i,a_{-i}^k}(w_{-i}^{k-1})\|_*^2\right], \quad (3)$$

*where $U_i := \max_{\sigma_i} \mathbb{E}\left[\phi_i^1(\sigma_i) - \phi_i^{T+1}(\sigma_i)\right] + \max_{\sigma_i} D(\sigma_i, \sigma_i^0)$, $\sigma^0, w^0$ are uniform and*

$$\phi_i^k(\sigma_i) = \alpha D(\sigma_i, \sigma_i^k) + (1-\alpha)/p \cdot D(\sigma_i, w_i^k) + (1-\alpha)D(\sigma_i^k, w_i^{k-1})$$
$$+ \tau\langle F_i(\sigma_{-i}^k) - F_i(w_{-i}^{k-1}), \sigma_i - \sigma_i^k\rangle. \quad (4)$$

Due to the space limit, the full proof of Theorem 1 is deferred to Appendix A.2. To obtain the main order of the above general upper bound, we first bound $U_i$, i.e., the first term in equation 3. According to the following Lemma 2, it is easy to see that $U_i$ is of order $\tilde{O}\left(1 + \tau + (1-\alpha)/p\right)$. When adopting hyper-parameters such that $\alpha \geq 1 - p$ and $\tau = O(1)$, we would have $U_i = \tilde{O}(1)$. Therefore, the key to bound the regret in Theorem 1 is to control the variance of the estimator, i.e., the third term in equation 3, which will be completed by our estimator introduced in Section 5.

**Lemma 2** (Bounds for $\phi_i^k$ defined in equation 4). *For any $k$ and $\sigma_i$, $\phi_i^k(\sigma_i) \geq -4\tau$. If the mirror map is $h_1$ and $w_i^k(a) = 1/A$, then $\phi_i^k(\sigma_i) \leq 4\tau + (1 + (1-\alpha)/p) \log A$. If the mirror map is $h_2$ and $w_i^k(a) = 1/A$, then $\phi_i^k(\sigma_i) \leq 4\tau + 2(1 + (1-\alpha)/p)$.*

## 5 LOW-VARIANCE MONTE-CARLO ESTIMATOR

In this section, we present our low-variance estimator for $F_i(\sigma_{-i}) - F_i(w_{-i})$ where $\sigma$ and $w$ are two strategy profiles. For convenience, we assume $\sigma \neq w$. It is standard to use importance sampling

to construct an unbiased estimator. Specifically, let $q_{-i}$ denote a distribution over $\mathcal{A}_{-i}$ such that $q_{-i}(a_{-i}) = 0$ only if $\sigma_{-i}(a_{-i}) = 0$ and $w_{-i}(a_{-i}) = 0$. Recall $F_{i,a_{-i}}(\sigma_{-i}) = F_i(a_{-i})\frac{\sigma_{-i}(a_{-i})}{q(a_{-i})}$. Clearly, we have $\mathbb{E}_{a_{-i} \sim q_{-i}}\left[F_{i,a_{-i}}(\sigma_{-i})\right] = \sum_{a_{-i}} F_i(a_{-i})\sigma_{-i}(a_{-i}) = F_i(\sigma_{-i})$.

Therefore, $F_{i,a_{-i}}(\sigma_{-i}) - F_{i,a_{-i}}(w_{-i})$ is an unbiased estimator for $F_i(\sigma_{-i}) - F_i(w_{-i})$ when $a_{-i} \sim q_{-i}$. However, for an arbitrary $q_{-i}$, the variance of the estimator can be very large. For example, let $q_{-i}$ denote the uniform distribution over $\mathcal{A}_{-i}$. Then, in the worst case, the variance of $F_{i,a_{-i}}(\sigma_{-i}) - F_{i,a_{-i}}(w_{-i})$ can be $A^N$. So we need to carefully design $q_{-i}$ to ensure low variance.

Note the variance $\mathbb{E}_{a_{-i} \sim q_{-i}}\left[\|F_{i,a_{-i}}(\sigma_{-i}) - F_{i,a_{-i}}(w_{-i})\|_\infty^2\right]$ [2] is upper bounded by $\sum_{a_{-i}} \|F_i(a_{-i})\|_\infty^2 (\sigma_{-i}(a_{-i}) - w_{-i}(a_{-i}))^2/q_{-i}(a_{-i})$. Intuitively, to control the variance, we should allocate a large probability mass to $a_{-i}$ where $(\sigma_{-i}(a_{-i}) - w_{-i}(a_{-i}))^2$ is large. With the observation that the difference between $\sigma_{-i}(a_{-i})$ and $w_{-i}(a_{-i})$ can be decomposed as $\sigma(a) - w(a) = \sum_{i=1}^N (\sigma_i(a_i) - w_i(a_i)) \prod_{x<i} \sigma_x(a_x) \prod_{y>i} w_y(a_y)$ (see Lemma 10 for more details), we propose to sample $a_{-i}$ according to the following distribution:

$$q_{-i}(a_{-i}) = \frac{1}{\sum_{j' \neq i} Z_{j'}} \sum_{j \neq i} |\sigma_j(a_j) - w_j(a_j)| \prod_{x<j, x \neq i} \sigma_x(a_x) \prod_{y>j, y\neq i} w_y(a_y), \qquad (5)$$

where $Z_j = \sum_{a_j} |\sigma_j(a_j) - w_j(a_j)|$. It is easy to verify that $q_{-i}(a_{-i}) \geq 0$ and $\sum_{a_{-i}} q_{-i}(a_{-i}) = 1$.

The following Algorithm 2 summarizes an efficient sampling procedure from $q_{-i}$ in equation 5 with polynomial time complexity $O((N-1)A)$. It takes two main steps to sample $a_{-i}$. Firstly, we sample index $j$ with probability $Z_j / \sum_{j' \neq i} Z_{j'}$ (Line 1). And then, we sample $a_{-i}$ with probability proportional to $|\sigma_j(a_j) - w_j(a_j)| \prod_{x<j, x \neq i} \sigma_x(a_x) \prod_{y>j, y\neq i} w_y(a_y)$ (Line 3-5).

---

**Algorithm 2** LVE$(i, \sigma, w)$

---

1: Compute $Z_j = \sum_{a_j} |\sigma_j(a_j) - w_j(a_j)|$ for $j \neq i$.
2: Sample $j$ with probability $\frac{Z_j}{\sum_{j' \neq i} Z_{j'}}$.
3: For $1 \leq j' < j, j' \neq i$, sample $a_{j'}$ according to $\sigma_{j'}$.
4: Sample $a_j$ with probability $\frac{|\sigma_j(a_j) - w_j(a_j)|}{\sum_{a_j'} |\sigma_j(a_j') - w_j(a_j')|}$.
5: For $j < j' \leq N, j' \neq i$, sample $a_{j'}$ according to $w_{j'}$.
6: **return** : $a_{-i}$

---

The following Lemma 3 provides an upper bound for the variance of the constructed estimator in equation 5. We believe this result is also of independent interest since it is quite general and may be used to control the variance of other stochastic algorithms for general games.

**Lemma 3** (Variance bound). *Sampling $a_{-i} \sim q_{-i}$ defined in equation 5 can be implemented by Algorithm 2. Moreover, we can upper bound its variance by*

$$\mathbb{E}_{a_{-i} \sim q_{-i}}\left[\|F_{i,a_{-i}}(\sigma_{-i}) - F_{i,a_{-i}}(w_{-i})\|_\infty^2\right] \leq (N-1) \sum_{j \neq i} \|\sigma_j - w_j\|_1^2. \qquad (6)$$

*Proof.* According to the definition of $q_{-i}$, the variance of the estimated loss is

$$\mathbb{E}_{a_{-i} \sim q_{-i}}\left[\|F_{i,a_{-i}}(\sigma_{-i}) - F_{i,a_{-i}}(w_{-i})\|_\infty^2\right] \leq \sum_{a_{-i}} \left|(\sigma_{-i}(a_{-i}) - w_{-i}(a_{-i}))\right|^2 / q_{-i}(a_{-i}).$$

Further, recall that the difference between $\sigma(a)$ and $w(a)$ can be decomposed as $\sum_{i=1}^N (\sigma_i(a_i) - w_i(a_i)) \prod_{x<i} \sigma_x(a_x) \prod_{y>i} w_y(a_y)$ in Lemma 10 (Please see appendix for the statement and proof),

---

[2]In Theorem 1, we consider the general dual norm $\| \cdot \|_*$. Here, we only upper bound $\| \cdot \|_\infty$ because when the mirror map is $h_1$ or $h_2$, the dual norm is $\| \cdot \|_\infty$ or $\| \cdot \|_2 \leq \sqrt{A}\| \cdot \|_\infty$, respectively.

it holds that

$$\sum_{a_{-i}} \left| \left( \sigma_{-i}(a_{-i}) - w_{-i}(a_{-i}) \right) \right|^2 / q_{-i}(a_{-i})$$

$$= \sum_{a_{-i}} \left| \left( \sum_{j \neq i} (\sigma_j(a_j) - w_j(a_j)) \prod_{x<j,x\neq i} \sigma_x(a_x) \prod_{y>j,y\neq i} w_y(a_y) \right) \right|^2 / q_{-i}(a_{-i})$$

$$\leq \sum_{a_{-i}} \left| \left( \sum_{i=1}^{N} |\sigma_i(a_i) - w_i(a_i)| \prod_{x<j,x\neq i} \sigma_x(a_x) \prod_{y>j,y\neq i} w_y(a_y) \right) \right|^2 / q_{-i}(a_{-i})$$

$$= \left( \sum_{j \neq i} Z_j \right) \sum_{a_{-i}} \left| \left( \sum_{i=1}^{N} |\sigma_i(a_i) - w_i(a_i)| \prod_{x<j,x\neq i} \sigma_x(a_x) \prod_{y>j,y\neq i} w_y(a_y) \right) \right|$$

$$= \left( \sum_{j \neq i} Z_j \right)^2 \leq (N-1) \sum_{j \neq i} \|\sigma_j - w_j\|_1^2 .$$

□

**Discussion on the optimality of LVE**    There are mainly two aspects to examine the optimality of a Monte-Carlo estimator: the variance and computational complexity.

1.    Variance:    It is obvious that the smallest variance is $(\sum_{a_{-i}} |\sigma_{-i}(a_{-i}) - w_{-i}(a_{-i})|)^2$ achieved by $q_{-i}^{\text{CJST19}}$ (CJST19).    Then, after noticing that $q_{-i}(a_{-i}) \geq q_{-i}^{\text{CJST19}}(a_{-i})(\sum_{a_{-i}} |\sigma_{-i}(a_{-i}) - w_{-i}(a_{-i})|)/\sum_{j \neq i} Z_j \geq q_{-i}^{\text{CJST19}}(a_{-i})/(N-1)$, the variance of our estimator is optimal within a multiplicative factor $N-1$ according to Lemma 4.

**Lemma 4.** *For any $\sigma$ and $w$, we have the variance of estimator with $q_{-i}$ from equation 5 is no more than $(N-1)(\sum_{a_{-i}} |\sigma_{-i}(a_{-i}) - w_{-i}(a_{-i})|)^2$.*

2. Computational complexity: The computational cost of our LVE is also optimal among estimators with low variance. Intuitively, the time complexity of LVE equals to access each entry of $\sigma_j^k$ and $w_j^{k-1}$ for constant times, and it is unlikely to be improved. The following lemma formally shows this.

**Lemma 5.** *We say $j$ is agnostic to $q'_{-i}$ if there are at least two entries of $\sigma_j$ are not accessed when computing $q'_{-i}$. Let $m$ denote the number of $j$ which is agnostic to $q'_{-i}$. Then, there exists $\sigma$ and $w$, the variance of the estimator is $\Omega(2^m)$.*

It is remarkable that though the estimator of (CJST19) achieves the smallest variance, its computational complexity is $O(|A|^{N-1})$ to access every configuration of $a_{-i}$ to compute $|\sigma_{-i}(a_{-i}) - w_{-i}(a_{-i})|$, which is impractical in general games with multiple players. Our LVE estimator simultaneously guarantees (near-)optimal variance and computational complexity.

## 6    REGRET UPPER BOUNDS AND TIME COMPLEXITY TO APPROXIMATE CCE

Now we are ready to provide our final guarantees to reach weak $\epsilon$-CCE which corresponds to social welfare and strong $\epsilon$-CCE which corresponds to individual loss. In addition, we also provide an adversarial regret bound of $O(\sqrt{T})$ to show the robustness of our algorithm.

### 6.1    SOCIAL WELFARE

We first consider the time complexity to reach a weak $\epsilon$-CCE. For this case, we take the mirror map $h_1(x) = \sum_{a=1}^{d} x(a) \log x(a)$ as an example. Recall that Lemma 1 shows that the time complexity to approximate CCE depends on both the regret and the per-round running time. So we first provide an upper bound for the regret $\sum_{i \in N} \max_{\sigma_i} \mathbb{E}\left[R_i(\sigma_i)\right]$ defined in weak $\varepsilon$-CCE.

**Theorem 2.** *Let hyper-parameters $\alpha = 1 - p$, $\tau = \sqrt{\gamma\alpha(1-\alpha)/2}/((N-1)\sqrt{1+\alpha\gamma})$, $\gamma \in (0, 1/2)$ be a constant and the mirror map be $h_1$. Then there exists a constant $C$ such that $\max_\sigma \mathbb{E}\left[\sum_{i=1}^{N} R_i(\sigma_i)\right] \leq CN^2 \log A/\sqrt{p}.$*

With the above upper bound, we can then guarantee the time complexity to reach a weak $\epsilon$-CCE.

**Corollary 1.** *With $p = NA/Cost$ and the hyper-parameters defined as Theorem 2, the time complexity of Algorithm 1 to reach a weak $\epsilon$-CCE is $O\left(Cost + N^{7/2}\sqrt{ACost}\log A/\epsilon\right)$.*

## 6.2 INDIVIDUAL REGRET

In this subsection, we provide the time complexity to reach a strong $\epsilon$-CCE. As previous analysis, we will also first give the upper bound for the individual regret. Here we take the mirror map $h_2(x) = \sum_{a=1}^{d} x^2(a)$ as an example.

Before providing the upper bound for the individual regret, we first present a useful lemma to bound the term $\mathbb{E}\left[\|\sigma_j^k - w_j^{k-1}\|_1^2\right]$ appearing in the general upper bound in Theorem 1.

**Lemma 6.** *If the hyper-parameter $\alpha = 1$, then $\mathbb{E}\left[\|\sigma_j^k - w_j^{k-1}\|_1^2\right] \leq 2\tau^2/p^2$.*

With Lemma 6, we now provide an upper bound on individual regret.

**Theorem 3.** *With $\tau = \left(A\left(2 + 1/\gamma\right)(N-1)^2 T/p^2\right)^{-1/4}$, $\alpha = 1$, $\gamma \in (0, 1/2)$ and mirror map $h_2$, we have $\max_{\sigma_i} \mathbb{E}\left[R_i(\sigma_i)\right] = O\left(A(N-1)^2 T/p^2\right)^{1/4}$ for any $i \in [N]$.*

The above individual regret upper bound in Theorem 3 implies the following time-complexity to reach a strong $\epsilon$-CCE.

**Corollary 2.** *With $p = NA/Cost$ and the hyper-parameters defined in Theorem 3, the running time to reach a strong $\epsilon$-CCE is $O\left(Cost + N^2 ACost^{2/3}/\epsilon^{4/3}\right)$.*

## 6.3 ADVERSARIAL REGRET

While this work mainly focuses on the case that every player adopts the same algorithm to optimize its own strategy, it is also interesting to consider the case where some players play adversarially, i.e., $\exists$ player $j$ who updates $\sigma_j^k$ and $w_j^{k-1}$ adversarially, to see if the algorithm is robust. According to Lemma 3, we have

$$\mathbb{E}_{a_{-i} \sim q_{-i}}\left[\|F_{i,a_{-i}}(\sigma_{-i}) - F_{i,a_{-i}}(w_{-i})\|_\infty^2\right] \leq (N-1)\sum_{j \neq i}\|\sigma_j - w_j\|_1^2 \leq N(N-1),$$

which also holds in the adversarial setting. Then, inserting this inequality into Theorem 1 and with standard derivation, we have an $O(\sqrt{T})$ adversarial regret which is formally presented below.

**Lemma 7.** *With the mirror map $h_2$, $\alpha = 1$, $\tau = N\sqrt{T}$ and $\gamma \in (0, 1/2)$, if player $i$ follows Algorithm 1, then for any $\sigma_j^k, w_j^{k-1}, j \neq i, k = 1, \cdots, T$, we have $\mathbb{E}\left[R_i(\sigma_i)\right] = O(N\sqrt{T})$.*

## 7 CONCLUSION

In this paper, we propose a stochastic version of OMD with variance reduction. Our algorithm extends prior works on variance-reduced stochastic algorithms from two-player zero-sum games to general games. The key innovation of this work is a low-variance Monte-Carlo estimator. In comparison with the prior estimator in (CJST19), our estimator is exponentially fast with a slightly larger variance.

There are several directions to extend our algorithm: Firstly, despite our algorithm enjoying an $O(1/\epsilon)$ convergence rate to weak-CCE, its convergence rate to strong CCE is $O(1/\epsilon^{4/3})$ which seems to be sub-optimal in comparison to the convergence rate in (DFG21) and (PSS21). Thus, developing stochastic algorithms with an $O(1/\epsilon)$ convergence rate to strong-CCE is an interesting direction; Secondly, we only consider normal-form games. However, it is more realistic to consider games with sequential structures, e.g., extensive-form games. We hope this work could be a starting point for developing more efficient stochastic algorithms for general games.

## ACKNOWLEDGEMENT

Shuai Li is supported by National Natural Science Foundation of China (92270201, 62006151, 62076161) and Shanghai Sailing Program. The underlining research carried out in this paper is not a part of any projects funded by the National Natural Science Foundation of China.

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

## A PROOF

Our regret bounds are derived from the following first-order optimality condition which is according to the update rule in Line 9 in Alg. 1. We have

$$0 \le \langle \nabla h(\sigma_i^{k+1}) - \nabla h(\tilde{\sigma}_i^{k+1}), \sigma_i - \sigma_i^{k+1} \rangle$$
$$= \langle \nabla h(\sigma_i^{k+1}) - \nabla h(\hat{\sigma}_i^k) + \tau(F_i(w_{-i}^k) + F_{i,a_{-i}^k}(\sigma_{-i}^k) - F_{i,a_{-i}^k}(w_{-i}^{k-1})), \sigma_i - \sigma_i^{k+1} \rangle. \quad (7)$$

Adding $\tau \langle F_i(\sigma_{-i}^{k+1}), \sigma_i^{k+1} - \sigma_i \rangle$ to both sides of equation 7, we immediately get an upper bound on the regret.

$$\tau \langle F_i(\sigma_{-i}^{k+1}), \sigma_i^{k+1} - \sigma_i \rangle \le \langle \nabla h(\sigma_i^{k+1}) - \nabla h(\hat{\sigma}_i^k), \sigma_i - \sigma_i^{k+1} \rangle$$
$$+ \tau \langle F_i(w_{-i}^k) + F_{i,a_{-i}^k}(\sigma_{-i}^k) - F_{i,a_{-i}^k}(w_{-i}^{k-1}) - F_i(\sigma_{-i}^{k+1}), \sigma_i - \sigma_i^{k+1} \rangle. \quad (8)$$

The rest of our proof starts from equation 8.

### A.1 USEFUL LEMMAS

The following lemmas are useful in the proof of Theorem 1 and Lemma 3.

**Lemma 8.** *Define* $\mathbb{E}_k[\cdot] = \mathbb{E}[\cdot | \sigma^k, w^{k-1}]$. *Recall* $\nabla h(\hat{\sigma}_i^k) = \alpha \nabla h(\sigma_i^k) + (1-\alpha) \nabla h(w_i^k)$ *and*

$$\phi_i^k(\sigma_i) = \alpha D(\sigma_i, \sigma_i^k) + \frac{1-\alpha}{p} D(\sigma_i, w_i^k) + (1-\alpha) D(\sigma_i^k, w_i^{k-1}) + \tau \langle F_i(\sigma_{-i}^k) - F_i(w_{-i}^{k-1}), \sigma_i - \sigma_i^{k+1} \rangle,$$

*we have*

$$\phi_i^k(\sigma_i) - \phi_i^{k+1}(\sigma_i)$$
$$= \langle \nabla h(\sigma_i^{k+1}) - \nabla h(\hat{\sigma}_i^k), \sigma_i - \sigma_i^{k+1} \rangle + \frac{1-\alpha}{p} \mathbb{E}_k \left[ D(\sigma_i, w_i^{k+1}) \right] + \alpha D(\sigma_i^{k+1}, \sigma_i^k) + (1-\alpha) D(\sigma_i^k, w_i^{k-1})$$
$$- \frac{1-\alpha}{p} D(\sigma_i, w_i^{k+1}) + \tau \langle F_i(\sigma_{-i}^k) - F_i(w_{-i}^{k-1}), \sigma_i - \sigma_i^k \rangle + \tau \langle F_i(w_{-i}^k) - F_i(\sigma_{-i}^{k+1}), \sigma_i - \sigma_i^{k+1} \rangle.$$

*Proof.* We can decompose $\langle \nabla h(\sigma_i^{k+1}) - \nabla h(\hat{\sigma}_i^k), \sigma_i - \sigma_i^{k+1} \rangle$ as follows.

$$\langle \nabla h(\sigma_i^{k+1}) - \nabla h(\hat{\sigma}_i^k), \sigma_i - \sigma_i^{k+1} \rangle$$
$$= \alpha \langle \nabla h(\sigma_i^{k+1}) - \nabla h(\sigma_i^k), \sigma_i - \sigma_i^{k+1} \rangle + (1-\alpha) \langle \nabla h(\sigma_i^{k+1}) - \nabla h(w_i^k), \sigma_i - \sigma_i^{k+1} \rangle$$
$$= \alpha \left( D(\sigma_i, \sigma_i^k) - D(\sigma_i^{k+1}, \sigma_i^k) - D(\sigma_i, \sigma_i^{k+1}) \right)$$
$$\quad + (1-\alpha) \left( D(\sigma_i, w_i^k) - D(\sigma_i^{k+1}, w_i^k) - D(\sigma_i, \sigma_i^{k+1}) \right)$$
$$= \alpha \left( D(\sigma_i, \sigma_i^k) - D(\sigma_i^{k+1}, \sigma_i^k) - D(\sigma_i, \sigma_i^{k+1}) \right)$$
$$\quad + (1-\alpha) \left( D(\sigma_i, w_i^k) - D(\sigma_i^{k+1}, w_i^k) - D(\sigma_i, \sigma_i^{k+1}) \right)$$
$$\quad + \frac{1-\alpha}{p} \mathbb{E}_k \left[ D(\sigma_i, w_i^{k+1}) \right] - \frac{1-\alpha}{p} \mathbb{E}_k \left[ D(\sigma_i, w_i^{k+1}) \right]$$
$$= \left( \alpha D(\sigma_i, \sigma_i^k) + \frac{1-\alpha}{p} D(\sigma_i, w_i^k) \right) - \left( \alpha D(\sigma_i, \sigma_i^{k+1}) + \frac{1-\alpha}{p} \mathbb{E}_k \left[ D(\sigma_i, w_i^{k+1}) \right] \right)$$
$$\quad - \alpha D(\sigma_i^{k+1}, \sigma_i^k) - (1-\alpha) D(\sigma_i^{k+1}, w_i^k)$$
$$= \left( \alpha D(\sigma_i, \sigma_i^k) + \frac{1-\alpha}{p} D(\sigma_i, w_i^k) + (1-\alpha) D(\sigma_i^k, w_i^{k-1}) \right)$$
$$\quad - \left( \alpha D(\sigma_i, \sigma_i^{k+1}) + \frac{1-\alpha}{p} \mathbb{E}_k \left[ D(\sigma_i, w_i^{k+1}) \right] - (1-\alpha) D(\sigma_i^{k+1}, w_i^k) \right)$$
$$\quad - \alpha D(\sigma_i^{k+1}, \sigma_i^k) - (1-\alpha) D(\sigma_i^k, w_i^{k-1}),$$

where the first equality is based on the fact that $\nabla h(\hat{\sigma}_i^k) = \alpha \nabla h(\sigma_i^k) + (1-\alpha)\nabla h(w_i^k)$, the second one is based on the definition of the Bregman divergence, the fourth one is according to the updating rule of $w_i^{k+1}$ (Line 10 in Alg. 1) and the third and the last equality holds obviously. Further with the definition of $\phi_i^k$, we complete the proof. $\qquad\square$

**Lemma 9** (Lemma 3.5 in (AM21)). *Let $\mathcal{F} = \{\mathcal{F}^k\}_{k\geq 0}$ be a filtration and $(u^k)$ be a stochastic process adapted to $\mathcal{F}$ with $\mathbb{E}[u^{k+1}|\mathcal{F}^k] = 0$. Then for any $x_0$ and any compact set $\mathcal{C}$,*

$$\mathbb{E}\left[\max_{x\in\mathcal{C}}\sum_{k=0}^{K-1}\langle u^{k+1}, x\rangle\right] \leq \max_{x\in\mathcal{C}} D(x, x^0) + \frac{1}{2}\sum_{k=0}^{K-1}\mathbb{E}\left[\|u^{k+1}\|_*^2\right].$$

**Lemma 10.** *For any $\sigma(a) = \prod_{j=1}^N \sigma_j(a_j), w(a) = \prod_{j=1}^N w_j(a_j)$, we have*

$$\sigma(a) - w(a) = \sum_{i=1}^N (\sigma_i(a_i) - w_i(a_i)) \prod_{x<i} \sigma_x(a_x) \prod_{y>i} w_y(a_y). \tag{9}$$

*Proof.* We prove this lemma by mathematical induction. The case when $N = 1$ holds obviously. Further, assume Lemma 10 holds for $N - 1$, then for $N$, it holds that

$$\prod_{j=1}^N \sigma_j(a_j) - \prod_{j=1}^N w_j(a_j)$$
$$= \prod_{j=1}^N \sigma_j(a_j) - w_N(a_N)\prod_{j=1}^{N-1}\sigma_j(a_j) + w_N(a_N)\prod_{j=1}^{N-1}\sigma_j(a_j) - \prod_{j=1}^N w_j(a_j)$$
$$= (\sigma_N(a_N) - w_N(a_N))\prod_{j=1}^{N-1}\sigma_j(a_j) + w_N(a_N)\left(\prod_{j=1}^{N-1}\sigma_j(a_j) - \prod_{j=1}^{N-1}w_j(a_j)\right)$$
$$= \sum_{i=1}^N (\sigma_i(a_i) - w_i(a_i))\prod_{x<i}\sigma_x(a_x)\prod_{y>i}w_y(a_y).$$

$\qquad\square$

## A.2 PROOF OF THEOREM 1

If $D(x, y) \geq \gamma\|x - y\|^2$. Summing equation 8 over $k = 1, \cdots, T$, applying Lemma 8 and taking expectation on both sides, we have

$$\max_{\sigma_i} \mathbb{E}\left[\sum_{k=1}^{T}\tau\langle F_i(\sigma_{-i}^{k+1}),\sigma_i^{k+1}-\sigma_i\rangle\right]$$

$$\overset{①}{\leq}\max_{\sigma_i}\mathbb{E}\left[\sum_{k=1}^{T}\left(\phi_i^k(\sigma_i)-\phi_i^{k+1}(\sigma_i)+\tau\langle F_{i,a_{-i}^k}(\sigma_{-i}^k)-F_{i,a_{-i}^k}(w_{-i}^{k-1}),\sigma_i-\sigma_i^{k+1}\rangle\right.\right.$$
$$-\tau\langle F_i(\sigma_{-i}^k)-F_i(w_{-i}^{k-1}),\sigma_i-\sigma_i^k\rangle-\left((1-\alpha)D(\sigma_i^k,w_i^{k-1})+\alpha D(\sigma_i^{k+1},\sigma_i^k)\right)$$
$$\left.\left.-\left(\frac{1-\alpha}{p}\mathbb{E}_k\left[D(\sigma_i,w_i^{k+1})\right]-\frac{1-\alpha}{p}D(\sigma_i,w_i^{k+1})\right)\right)\right]$$

$$\leq\max_{\sigma_i}\mathbb{E}\left[\phi_i^1(\sigma_i)-\phi_i^{T+1}(\sigma_i)\right]$$
$$+\max_{\sigma_i}\mathbb{E}\left[\sum_{k=1}^{T}\tau\langle F_{i,a_{-i}^k}(\sigma_{-i}^k)-F_{i,a_{-i}^k}(w_{-i}^{k-1})-\left(F_i(\sigma_{-i}^k)-F_i(w_{-i}^{k-1})\right),\sigma_i-\sigma_i^k\rangle\right]$$
$$+\max_{\sigma_i}\mathbb{E}\left[\sum_{k=1}^{T}\left(\tau\langle F_{i,a_{-i}^k}(\sigma_{-i}^k)-F_{i,a_{-i}^k}(w_{-i}^{k-1}),\sigma_i^k-\sigma_i^{k+1}\rangle-\left((1-\alpha)D(\sigma_i^k,w_i^{k-1})+\alpha D(\sigma_i^{k+1},\sigma_i^k)\right)\right)\right]$$
$$+\max_{\sigma_i}\mathbb{E}\left[\sum_{k=1}^{T}-\left(\frac{1-\alpha}{p}\mathbb{E}_k\left[D(\sigma_i,w_i^{k+1})\right]-\frac{1-\alpha}{p}D(\sigma_i,w_i^{k+1})\right)\right]$$

$$\overset{②}{\leq}\max_{\sigma_i}\mathbb{E}\left[\phi_i^1(\sigma_i)-\phi_i^{T+1}(\sigma_i)\right]$$
$$+\max_{\sigma_i}\mathbb{E}\left[\sum_{k=1}^{T}\tau\langle F_{i,a_{-i}^k}(\sigma_{-i}^k)-F_{i,a_{-i}^k}(w_{-i}^{k-1})-\left(F_i(\sigma_{-i}^k)-F_i(w_{-i}^{k-1})\right),\sigma_i-\sigma_i^k\rangle\right]$$
$$+\mathbb{E}\left[\sum_{k=1}^{T}\left(\frac{\tau^2}{\alpha\gamma}\|F_{i,a_{-i}^k}(\sigma_{-i}^k)-F_{i,a_{-i}^k}(w_{-i}^{k-1})\|_*^2+\alpha\gamma\|\sigma_i^{k+1}-\sigma_i^k\|^2-\left((1-\alpha)D(\sigma_i^k,w_i^{k-1})+\alpha D(\sigma_i^{k+1},\sigma_i^k)\right)\right)\right]$$

$$\leq\max_{\sigma_i}\mathbb{E}\left[\phi_i^1(\sigma_i)-\phi_i^{T+1}(\sigma_i)\right]$$
$$+\max_{\sigma_i}\mathbb{E}\left[\sum_{k=1}^{T}\tau\langle F_{i,a_{-i}^k}(\sigma_{-i}^k)-F_{i,a_{-i}^k}(w_{-i}^{k-1})-(F_i(\sigma_{-i}^k)-F_i(w_{-i}^{k-1})),\sigma_i-\sigma_i^k\rangle\right]$$
$$+\mathbb{E}\left[\sum_{k=1}^{T}\left(\frac{\tau^2}{\alpha\gamma}\|F_{i,a_{-i}^k}(\sigma_{-i}^k)-F_{i,a_{-i}^k}(w_{-i}^{k-1})\|_*^2-(1-\alpha)D(\sigma_i^k,w_i^{k-1})\right)\right]$$

$$\overset{③}{\leq}\max_{\sigma_i}\mathbb{E}\left[\phi_i^1(\sigma_i)-\phi_i^{T+1}(\sigma_i)\right]+\max_{\sigma_i}D(\sigma_i,\sigma_i^0)+\tau^2\mathbb{E}\left[\sum_{k=1}^{T}\|F_{i,a_{-i}^k}(\sigma_{-i}^k)-F_{i,a_{-i}^k}(w_{-i}^{k-1})\|_*^2\right]$$
$$+\mathbb{E}\left[\sum_{k=1}^{T}\frac{\tau^2}{\alpha\gamma}\|F_{i,a_{-i}^k}(\sigma_{-i}^k)-F_{i,a_{-i}^k}(w_{-i}^{k-1})\|_*^2-(1-\alpha)D(\sigma_i^k,w_i^{k-1})\right]$$

$$\leq\max_{\sigma_i}\mathbb{E}\left[\phi_i^1(\sigma_i)-\phi_i^{T+1}(\sigma_i)\right]+\max_{\sigma_i}D(\sigma_i,\sigma_i^0)$$
$$-(1-\alpha)\mathbb{E}\left[\sum_{k=1}^{T}D(\sigma_i^k,w_i^{k-1})\right]+\tau^2\left(1+\frac{1}{\alpha\gamma}\right)\mathbb{E}\left[\sum_{k=1}^{T}\|F_{i,a_{-i}^k}(\sigma_{-i}^k)-F_{i,a_{-i}^k}(w_{-i}^{k-1})\|_*^2\right],$$

where ①  is based on Lemma 8; ② is according to Young's inequality and ③ is derived from Lemma 9.

## B    MISSING PROOFS

*Proof of Lemma 2.*  The lower bound follows directly from the non-negativity of Bregman divergence $D(x, y)$. The upper bound of $\phi_i^k$ for $h_1$ is based on the fact that $\max_{\sigma_i} D(\sigma_i, w_i^k) = \log A$ when $w_i^K(a) = 1/A$. And the case of $h_2$ can also be proved in a similar way.  □

*Proof of Theorem 2.*  When the mirror map is $h_1(x) = \sum_{a=1}^d x(a) \log x(a)$, it is known that $D(x, y) \geq \|x - y\|_1^2$ and the dual norm is $\| \cdot \|_\infty$. Thus a direct combination of Theorem 1 and Lemma 3 yields that

$$
\tau \max_{\sigma_i} \mathbb{E}\left[R_i(\sigma_i)\right] \leq U_i - (1 - \alpha)\mathbb{E}\left[\sum_{k=1}^T D(\sigma_i^k, w_i^{k-1})\right]
$$

$$
+ (N - 1)\tau^2 \left(1 + \frac{1}{\alpha\gamma}\right) \mathbb{E}\left[\sum_{k=1}^T \sum_{j \neq i} \|\sigma_j^k - w_j^{k-1}\|_1^2\right]. \tag{10}
$$

Further, summing equation 10 over $i = 1, \cdots, N$, we have

$$
\sum_{i=1}^N \tau \max_{\sigma_i} \mathbb{E}\left[R_i(\sigma_i)\right] \leq \sum_{i=1}^N U_i - \frac{1 - \alpha}{2} \sum_{i=1}^N \mathbb{E}\left[\sum_{k=1}^T \|\sigma_i^k - w_i^{k-1}\|_1^2\right]
$$

$$
+ (N - 1)^2 \tau^2 \left(1 + \frac{1}{\alpha\gamma}\right) \mathbb{E}\left[\sum_{k=1}^T \sum_{i=1}^N \|\sigma_i^k - w_i^{k-1}\|_1^2\right] = \sum_{i=1}^N U_i.
$$

Recall that Lemma 2 shows $U_i$ is of order $O\left((1 - \alpha) \log A/p + \tau\right)$ when adopting mirror map $h_1$. By choosing $\alpha = 1 - p$ and $\gamma \in (0, 1/2)$, we can see that there exist constants $C, C'$ such that $\sum_{i=1}^N \max_{\sigma_i} \mathbb{E}\left[R_i(\sigma_i)\right] \leq C'N \left(1 + \log A/\tau\right) \leq CN^2 \log A/\sqrt{p}$.  □

*Proof of Corollary 1.*  According to Theorem 2, to arrive a weak $\epsilon$-CCE, we need to run Algorithm 1 for $T = \left(CN^2 \log A\right)/\left(\epsilon\sqrt{p}\right)$ rounds. Recall that the per round time complexity is $O(pN\text{Cost} + N^2 A)$. Then the total running time is $O(\text{Cost} + (pN\text{Cost} + N^2 A)N^2 \log A/(\epsilon\sqrt{p}) = O(\text{Cost} + N^{7/2}\sqrt{A\text{Cost}} \log A/\epsilon)$ with $p = NA/\text{Cost}$.  □

*Proof of Lemma 6.*  With $\alpha = 1$, it holds that

$$
\mathbb{E}\left[\|\sigma_j^k - w_j^{k-1}\|_1^2\right] = \mathbb{E}\left[\sum_{t=1}^{k-1} \text{Prob}[w_j^{k-1} = \sigma_j^t]\|\sigma_j^k - \sigma_j^t\|_1^2\right]
$$

$$
= \mathbb{E}\left[\sum_{t=1}^{k-1} p(1 - p)^{k-1-t}\|\sigma_j^k - \sigma_j^t\|_1^2\right]
$$

$$
= \sum_{t=1}^{k-1} p(1 - p)^{k-1-t}(k - t)^2\tau^2 \leq 2\tau^2/p^2.
$$

□

*Proof of Theorem 3.*  When the mirror map is $h_2(x) = \frac{1}{2}\sum_{a=1}^d x^2(a)$, it is known that is $D(x, y) = \frac{1}{2}\|x - y\|_2^2$ and the dual norm is $\| \cdot \|_2$. Combining the results of Theorem 1, Lemma 3 , Lemma 6, and the fact that $\|F_{i,a_{-i}^k}(\sigma_{-i}^k) - F_{i,a_{-i}^k}(w_{-i}^{k-1})\|_2 \leq \sqrt{A}\|F_{i,a_{-i}^k}(\sigma_{-i}^k) - F_{i,a_{-i}^k}(w_{-i}^{k-1})\|_\infty$, we have

$$
\max_{\sigma_i} \mathbb{E}\left[R_i(\sigma_i)\right] \leq U_i/\tau + 2\tau^3 A \left(2 + 1/\gamma\right)(N - 1)^2 T/p^2.
$$

And Lemma 2 implies that $U_i = O(1 + \tau)$ when $\alpha = 1$. Let $\tau = \left( A\left(2 + 1/\gamma\right)(N-1)^2 T/p^2\right)^{-1/4}$, we have

$$\max_{\sigma_i} \mathbb{E}\left[R_i(\sigma_i)\right] = O\left(A\left(2 + 1/\gamma\right)(N-1)^2 T/p^2\right)^{1/4}.$$

$\square$

*Proof of Corollary 2.* According to the upper bound in Theorem 3, it needs $T = O((A(N-1)/p)^{1/3}\epsilon^{-4/3})$ rounds to reach an expected strong $\epsilon$-CCE. Recall that the per round time complexity is $O(pN\text{Cost} + N^2 A)$. Thus the expected running time is

$$O\left(\text{Cost} + \left(A(N-1)/p\right)^{1/3}\epsilon^{-4/3}\left(pN\text{Cost} + N^2 A\right)\right).$$

By replacing $p$ with $NA/\text{Cost}$, we can conclude that the running time to reach an expected strong $\epsilon$-CCE is $O\left(\text{Cost} + N^{7/3}A^{2/3}\text{Cost}^{2/3}/\epsilon^{4/3}\right)$.

$\square$

*Proof of Lemma 4.* Recall

$$q_{-i}(a_{-i}) = \frac{1}{\sum_{j' \neq i} Z_{j'}} \sum_{j \neq i} |\sigma_j(a_j) - w_j(a_j)| \prod_{x < j, x \neq i} \sigma_x(a_x) \prod_{y > j, y \neq i} w_y(a_y)$$

and

$$q_{-i}^{\text{CJST19}}(a_{-i}) = |\sigma_{-i}(a_{-i}) - w_{-i}(a_{-i})| / \sum_{a'_{-i}} |\sigma_{-i}(a'_{-i}) - w_{-i}(a'_{-i})|.$$

We have

$$\frac{q_{-i}(a_{-i})}{q_{-i}^{\text{CJST19}}(a_{-i})} = \frac{\sum_{a'_{-i}} |\sigma_{-i}(a'_{-i}) - w_{-i}(a'_{-i})|}{\sum_{j' \neq i} Z_{j'}} \frac{\sum_{j \neq i} |\sigma_j(a_j) - w_j(a_j)| \prod_{x < j, x \neq i} \sigma_x(a_x) \prod_{y > j, y \neq i} w_y(a_y)}{|\sigma_{-i}(a_{-i}) - w_{-i}(a_{-i})|}$$

$$\geq \frac{\sum_{a'_{-i}} |\sigma_{-i}(a'_{-i}) - w_{-i}(a'_{-i})|}{\sum_{j' \neq i} Z_{j'}} \frac{|\sum_{j \neq i} \sigma_j(a_j) - w_j(a_j) \prod_{x < j, x \neq i} \sigma_x(a_x) \prod_{y > j, y \neq i} w_y(a_y)|}{|\sigma_{-i}(a_{-i}) - w_{-i}(a_{-i})|}$$

$$= \frac{\sum_{a'_{-i}} |\sigma_{-i}(a'_{-i}) - w_{-i}(a'_{-i})|}{\sum_{j' \neq i} Z_{j'}},$$

where the last equality is according to Lemma 10. Further, let $a_{-i,-j}$ denote the action profile after removing $a_i$ and $a_j$, we have

$$\sum_{j \neq i} Z_j = \sum_{j \neq i} \sum_{a_j} |\sigma_j(a_j) - w_j(a_j)|$$

$$= \sum_{j \neq i} \sum_{a_j} |\sum_{a_{-i,-j}} (\sigma_{-i}(a_{-i,-j}, a_j) - w_{-i}(a_{-i,-j}, a_j))|$$

$$\leq \sum_{j \neq i} \sum_{a_j} \sum_{a_{-i,-j}} |(\sigma_{-i}(a_{-i,-j}, a_j) - w_{-i}(a_{-i,-j}, a_j))|$$

$$= \sum_{j \neq i} \sum_{a_{-i}} |(\sigma_{-i}(a_{-i}) - w_{-i}(a_{-i}))|$$

$$= (N-1) \sum_{a_{-i}} |(\sigma_{-i}(a_{-i}) - w_{-i}(a_{-i}))|.$$

Therefore, we have $\frac{q_{-i}(a_{-i})}{q_{-i}^{\text{CJST19}}(a_{-i})} \geq 1/(N-1)$. And the variance of our LVE estimator can be bounded as

$$\sum_{a_{-i}} \left|(\sigma_{-i}(a_{-i}) - w_{-i}(a_{-i}))\right|^2 / q_{-i}(a_{-i})$$

$$\leq (N-1) \sum_{a_{-i}} \left|(\sigma_{-i}(a_{-i}) - w_{-i}(a_{-i}))\right|^2 / q_{-i}^{\text{CJLS19}}(a_{-i}) = (N-1)(\sum_{a_{-i}} |\sigma_{-i}(a_{-i}) - w_{-i}(a_{-i})|)^2.$$

We finish the proof.

$\square$

*Proof of Lemma 5.* Let $AG$ denote the set of agnostic $j$. If $j$ is agnostic to $q'_{-i}$, we assume $\sigma_j(a_j) = 0$ for all $a_j$ which has been accessed by $q'_{-i}$. We further assume $\sigma_{j'} = w_{j'}$ for $j' \notin AG$. Then we can construct a equivalent game $G$ with $|AG| + 1 = m + 1$ players and $|\mathcal{A}_{-i}| = 2^m$. Moreover, $q'_{-i}$ does not know any entries of $\sigma_{-i}$. Then for $w_{-i}(a_{-i}) = 1/2^m$ and $q'_{-i}$, , we have

$$\max_{\sigma_{-i}} \sum_{a_{-i}} \left| \left( \sigma_{-i}(a_{-i}) - w_{-i}(a_{-i}) \right) \right|^2 / q_{-i}(a_{-i})$$

$$\geq \max_{a_{-i}} \left| \left( 1 - w_{-i}(a_{-i}) \right) \right|^2 / q_{-i}(a_{-i}) + \sum_{a'_{-i} \neq a_{-i}} \frac{w^2_{-i}(a'_{-i})}{q_{-i}(a'_{-i})}$$

$$= \max_{a_{-i}} \frac{1}{q_{-i}(a_{-i})} - 2 \frac{w_{-i}(a_{-i})}{q_{-i}(a_{-i})} + \sum_{a'_{-i}} \frac{w^2_{-i}(a'_{-i})}{q_{-i}(a'_{-i})}$$

$$\geq 2^m - 2 \,.$$

$\square$

