# OpenReview forum: "Stochastic No-regret Learning for General Games with Variance Reduction"
_ICLR.cc/2023/Conference — ICLR 2023 poster_

### Official Review · Reviewer_ZKef · 2022-10-23

**Confidence:** 3
**Correctness:** 4
**Technical Novelty And Significance:** 3
**Empirical Novelty And Significance:** Not applicable
**Recommendation:** 8

**Clarity, Quality, Novelty And Reproducibility:**

Although the notation is heavy, I think this is almost unavoidable for studying multi-agent games. Besides that the paper is well-written and explains its motivation and makes clear comparison and literature review of prior work.

While following the general framework of variance-reduced methods in prior work, the idea of constructing low-variance computationally-efficient gradient estimator for the OMD step is interesting and novel. I think this may have a positive impact for studying general-sum games more broadly.

**Strength And Weaknesses:**

I think the paper studies a natural and important problem, the presentation is clear and the results look sound to me. I have greatly enjoyed reading this paper. I only have a few minor comments:

1) When presenting the table for comparison in time complexities, it may be helpful to highlight in which regime the first time in your complexity bound will directly dominate? And for what threshold of cost will your method offer improvement over prior ones? It may also be helpful to r remark that your method actually recovers two-player zero-sum case in prior work. Out of curiosity, I also wonder would a purely stochastic method (without variance reduction) work and be favorable here?

2) In section 5, it may be helpful (if you know about it) to remark a bit on the optimality of your constructed estimator, in terms of variance and computational complexity.


**Summary Of The Paper:**

The paper studied a notion of variance-reduced stochastic OMD for multi-agent general-sum standard games with an improved convergence speed over prior stochastic methods. The methods show improved convergence complexity over deterministic algorithms, especially when computing the full loss vector for each player is expensive. The method is a nontrivial generalization over the previous two-player zero-sum variance-reduced methods which uses Monte-Carlo estimator to deal with the challenge in high-variance of previous estimators due to the potentially large number of players.

**Summary Of The Review:**

I think this is a technically nice paper that studies an important problem for stochastic OMD for multi-agent games. The authors clearly state their contribution and point out the future work in their main paper. Due to these reasons I recommend acceptance of the paper.

---

> ### Author Response · Authors · 2022-11-09
> **Response to reviewer ZKef**
>
> Thank you for your comments and suggestions. Our response is stated below.
>
> > When presenting the table for comparison in time complexities, it may be helpful to highlight in which regime the first time in your complexity bound will directly dominate? And for what threshold of cost will your method offer improvement over prior ones?
>
> Thanks for the suggestion. Our result improves previous works for weak $\epsilon$-CCE when Cost $\geq NA$ and for strong $\epsilon$-CCE when Cost $\ge NA^2/\epsilon$. We have added the comparison in the revision.
>
> > Out of curiosity, I also wonder would a purely stochastic method (without variance reduction) work and be favorable here?
>
> Thanks. Actually, we fail to come up with a game where a purely stochastic methods are preferred.
>
> > In section 5, it may be helpful (if you know about it) to remark a bit on the optimality of your constructed estimator, in terms of variance and computational complexity.
>
> Thank you for the suggestion on discussing the optimality. We have added a paragraph in the revision to discuss optimality. And actually, our algorithm is (near)-optimal in both variance and time complexity in the following sense:
>
> # Optimality of LVE
>  The following discussion on optimality have been added in the revision and you can find the proof of Lemma 4 and Lemma 5 in the updated appendix.
>
>  Variance: It is obvious that the smallest variance of an estimator is $(\sum_{a_{-i}}|\sigma_{-i}(a_{-i})-w_{-i}(a_{-i})|)^2$ achieved by $q_{-i}^{\text{CJST19}}$. And  the variance of our estimator is optimal within a multiplicative factor of $N-1$ according to the following lemma:
>
> Lemma 4:
>  For any $\sigma$ and $w$, we have the variance of estimator with $q_{-i}$ from equation 7 is no more than $(N-1)(\sum_{a_{-i}}|\sigma_{-i}(a_{-i})-w_{-i}(a_{-i})|)^2$.
>
> Time complexity:  the computational cost of LVE is also optimal among estimators with low variance. Intuitively, the time complexity of LVE equals to access each entry of $\sigma_j^k$ and $w_j^{k-1}$ for constant times, and there is unlikely to make improvements on the time complexity. Formally, we provide a lemma as follows:
>
> Lemma 5:
>  We say $j$ is agnostic to $q_{-i}$ if there are at least two entries of $\sigma_j$ are not accessed when computing $q_{-i}$. Let $m$ denote the number of $j$ which is agnostic to $q_{-i}$. Then, there exists $\sigma$ and $w$, the variance of the estimator is $\Omega(2^{m})$.

---

> > ### Comment · Reviewer_ZKef · 2022-12-06
> > **Dear authors**
> >
> > Thanks, the update looks great to me and answers my question.
> >
> > One small follow-up question I have is, I am not fully understanding why "Actually, we fail to come up with a game where a purely stochastic methods are preferred." In fact I thought based on the bounds, for any games as long as the player number is big enough (N), then a purely stochastic method may be favorable? By fail to come up, do you mean theoretically or empirically?

---

> > > ### Author Response · Authors · 2022-12-06
> > > **Response to the follow-up question**
> > >
> > > Thanks for your follow-up question.
> > >
> > > We considered the case $\epsilon\rightarrow 0$. Therefore, for any game, pure stochastic algorithms are not preferred if $\epsilon$ is small enough, as they converge in a speed of $1/\sqrt{\epsilon}$.
> > >
> > > The case, where $N$ is large while $\epsilon$ is not small enough, is also very interesting. We will discuss this case in the revision.

---

### Official Review · Reviewer_nWqn · 2022-10-26

**Confidence:** 4
**Correctness:** 4
**Technical Novelty And Significance:** 3
**Empirical Novelty And Significance:** Not applicable
**Recommendation:** 8

**Clarity, Quality, Novelty And Reproducibility:**

The paper is mostly clear, most of the ideas and algorithms are extensions of papers (Carmon et al. 2019) and (Alacaoglu and Malitsky 2021), but need to be adapted to this setting and introduces a new estimator for sampling and computing the loss vector.

**Strength And Weaknesses:**

Strengths:
This paper extends the idea of using stochastic methods for learning the CCE in normal form general games and show convergence in the expected sense to a weak $\varepsilon$-CCE and strong  $\varepsilon$-CCE This requires a new low-variance estimator that is different from (Alacaoglu and Malitsky 2021)  and (Carmon et al. 2019). This matches the time-complexity in (Alacaoglu and Malitsky 2021)  and (Carmon et al. 2019), when its a two player zero-sum game and when dealing with $\varepsilon$-Nash equilibrium. For general games the time complexity for convergence to a weak $\varepsilon$-CCE is arguably claimed to be faster than (Daskalakis, Fishelson and Golowich 21).

Weaknesses:
They mainly analyze the expected regret with a stochastic algorithm, whilst (Daskalakis, Fishelson and Golowich 21) analyze the actual regret, albeit they have a deterministic algorithm. In my view, the time to converge may not be directly comparable as it is currently done in Table 1. However, one can see this is an extension to papers like (Alacaoglu and Malitsky 2021) and (Carmon et al. 2019), which is focused on the general-sum game setting.

In addition, in the algorithm, all the players seem to have a shared random bit. Do the authors think it is possible to overcome this?

One minor comment: In Line 6 of the Algorithm 1, it says compute $\hat{\sigma}^{k}_i$ such that... But nowhere, is the actual $\hat{\sigma}^{k}_i$ used and all the proofs mainly require the gradient, which is the term on the right hand side. I think it might be better to say that we compute the gradient of h, than the actual estimate $\hat{\sigma}^{k}_i$.

**Summary Of The Paper:**

The authors in this paper provide an algorithm which is a stochastic variant of OMD, that converges in the expected sense to a weak and strong $\varepsilon$-CCE. This is done by bounding the expected regret of the algorithm by, extending the
theoretical framework of analyzing regret bounds of stochastic OMD in (Alacaoglu and Malitsky 2021)  from two-player
zero-sum games to general sum games. In addition, they propose a low-variance Monte-Carlo estimator
for general sum games and the computational complexity of this estimator is only $O(A(N-1))$, where $A$ is the number of actions and $N$ is the number of players in the game, whereas previous works (Carmon et al. 2019) and (Alacaoglu and Malitsky 2021) analyze the stochastic variant of OMD for two player zero-sum games.


**Summary Of The Review:**

Overall, I think the paper tries to address an interesting direction to understand stochastic variants of OMD algorithms for general sum-games and its convergence to CCE's.

---

> ### Author Response · Authors · 2022-11-09
> **Response to reviewer nWqn**
>
> Thanks for your comments. Our response to your concerns is as follows.
>
> > They mainly analyze the expected regret with a stochastic algorithm, whilst (Daskalakis, Fishelson and Golowich 21) analyze the actual regret, albeit they have a deterministic algorithm. In my view, the time to converge may not be directly comparable as it is currently done in Table 1.
>
> Thank you for your comments. We agree that we used the expected time complexity for stochastic algorithms in Table 1, and we have highlighted the definition of time complexity for random algorithms in caption. However, it seems to be no better choice than directly comparing time complexity if one wants to show the advantage of random algorithms over deterministic algorithms. Such dilemma have been presented in many other problems. For example, people may also have to directly compare the time complexity between gradient descent and stochastic gradient descent.
>
> > In addition, in the algorithm, all the players seem to have a shared random bit. Do the authors think it is possible to overcome this?
>
> Thanks for your question on the shared randomness. Actually, as a replacement of the shared random variable $u\in [0,1]$, each player can sample a private $u_i\in [0,1]$ and updates its own strategy accordingly, and the analysis on the convergence rate in Theorem 1, Lemma 3, Theorem 2 and Theorem 3 remain the same.
>
> We use the shared random bit based on the considerition on reducing the per round time complexity. More specifically, as we have analysed in section 4 (the paragraph after "Estimated Loss"), we need to recompute $F_i(w_{-i}^k)$ once some player $j\neq i$ updated its "snapshot strategy" $w_j^k$. Without a shared random bit, the frequency of updating the loss is $N-1$ times higher than the case of shared randomness.
>
> As for other minor issues, we have addressed in the revision according to your comments.

---

> > ### Comment · Reviewer_nWqn · 2022-12-07
> > **After the rebuttal**
> >
> > After reading the rebuttal and the other reviews, I am happy with the changes being made in the paper. I will increase my score to 8.

---

### Official Review · Reviewer_pnEU · 2022-11-02

**Confidence:** 3
**Correctness:** 4
**Technical Novelty And Significance:** 3
**Empirical Novelty And Significance:** Not applicable
**Recommendation:** 8

**Clarity, Quality, Novelty And Reproducibility:**

The paper is mostly clear with novel algorithms that can be interest to the community.

**Strength And Weaknesses:**

**Strength**

The paper brings variance reduction into general-sum games and shows promising results in computing a CCE more efficiently. This seems to be a worthful attempt on the road to addressing more complicated games in a efficient manner by incorporating stochasticity.

The main algorithm construction and the design choices are clearly explained, modulo some minor points that may hinder the understanding (as listed below).

**Weakness**

I don't think there are any major flaws. Here are a few typos and minor remarks that I think should be addressed.
1. In the caption of Table 1, there are two phrases that begin with "For stochastic algorithms". The first one should be removed. Moreover, in the second one the first expected should be removed (i.e. the time complexity is the ~~expected~~ running time of achieving an expected approximation error).
2. The authors mention that weak $\epsilon$-CCE is related to social welfare without further explanation. Some clarification may be needed here. As far as I am aware, this does not apply to all the general-sum games but only to those that satisfy a certain 'smoothness' condition.
3. While the algorithm stated in the pseudo-code is correct, equation (1) seems to be wrong. In particular, the term $\sigma_i^k-\tau(\ldots)$ mixes quantities that live in the primal (players' strategies) and live in the dual (gradients).
4. At the beginning of Section 4: "where $a_{-i}\in\mathcal{A}_{-i}$ denote**s** a random variable ..." (add s)
5. LVE used in Algorithm 1 is not yet defined at this point.
6. It seems that $w^0_i$ and $\sigma^0_i$ used in the definition of $U_i$ are never defined (I suppose they are uniform distribution over the simplex).
7. On the third line of page 6, the phrase "sampled from $q_{-i}$" seems to be redundant.

**Summary Of The Paper:**

This paper studies regret minimization in general-sum finite games. Such guarantees directly translate into the ones on convergence rate to approximated coarse correlated equilibrium. The main contribution is a stochastic version of the optimistic mirror descent algorithm that allows to achieve smaller regret (in expectation) within a given computational budget. In particular, a SVRG-like variance reduction mechanism along with a tailored sampling distribution are the key components that enable the improvement.

**Summary Of The Review:**

Overall, I believe both the variance-reduction and the sampling algorithms are valuable (in terms of novelty, the former is a rather straightforward generalization from the two-player case while the later requires non-trivial modification). I thus recommend acceptance of the paper.

---

> ### Author Response · Authors · 2022-11-09
> **Response to reviewer pnEU**
>
> Thanks for you detailed comments. We have addressed the typos in 1,3,4,5,6 and 7 in the revision and carefully polished the draft. And according to your suggestion in 2, we have added a discussion about the relationship between sum of regrets and social welfare in smooth games in the revision.

---

### Official Review · Reviewer_YwXx · 2022-11-03

**Confidence:** 4
**Clarity, Quality, Novelty And Reproducibility:** See below
**Correctness:** 4
**Technical Novelty And Significance:** 3
**Empirical Novelty And Significance:** 3
**Recommendation:** 6

**Details Of Ethics Concerns:**

Non-applicable

**Strength And Weaknesses:**

Strengths:
1) Introduction of a new variance reduction method (LVE) resolving the curse of dimensionality of Carmon's et al 19 estimator
2) Leveraging SVRG techniques for VI and EG to describe an Optimistic Var Reduction Stochastic Mirror Descent method.
3) Better bounds for the regret for each player.
4) General Functions for simplex functions.

Please, I request from the authors for each of the weakness prescribed below to address it as a statement/question that necessates clarification
Weaknesses:
1) It was not clear to me what are the guarantees for the expected regret of the sum of all players and what is the statement for the pseudo-regret of the sum of all players. Can you please clarify
2) Obtaining convergence to a weak CCE via a pseudo-regret has been typically described in the literature? Is there any novelty in this point? I assume no, the ingenuity starts with the variance reduction methods, correct?
3) The authors proposed LVE instead of classical IWE with exploration term. Can the authors explain the details of their estimator. It would help if they provide some extended answer. how they derive this estimator. Is it something standard in the literature?
4)There is another point which was not clear, the regret statements are correct if all the players employ the same algorithm? Do we have the adversarial 1/\sqrt{T} case, if one of the players plays adversarially/stochastically?

**Summary Of The Paper:**

This work leverages recent techniques for variance reduction in solving variational inequalities to obtain better bounds simultaneously in individual and sum of all players' regret for a general type of games.
More precisely, in their model, the authors endowed continuous functions as utility functions (not necessarily convex loss/concave gain) but the strategy of each player belongs to the simplex of a discrete action set, which is actually equivalent to normal form games. (Please correct me if I am wrong)

**Summary Of The Review:**

I had the chance to review the paper in this initial phase as emergency reviewer.
Having said that, I am very open to the discussion to be sure that I have not misunderstood anything in the paper. I am very familiar with the work of AM21 and the connections of Regret and CCE. Someone could say that there is at a first glance a plug-and-play situation. However, it seems that a new MC estimator is also a necessary contribution to go from minimization of one agent (AM21) to our CCE (collaborative regret minimization).

I have set a series of questions which if they will be addressed, I will be convinced that the paper deserves a clear accept. My positive score should be explained as my positive propensity for the papers' result.

---

> ### Author Response · Authors · 2022-11-09
> **Response to reviewer YwXx**
>
> We thank reviewer YwXx for the emergent review. Below we address you concerns.
>
> >It was not clear to me what are the guarantees for the expected regret of the sum of all players and what is the statement for the pseudo-regret of the sum of all players. Can you please clarify
>
> Thanks for the question. As said by reviewer pnEU, minimizing the summation of regret is closely related to a large class of games known as smooth games [1,2]. In a nutshell, let $F_{i}^k$ denote the sum of rewards received by $i$ in round $k$ and $F^k=\sum_i w_i^k$. Assume $F^*=\max_{\sigma}\sum_i F_i(\sigma)$ denote the optimal welfare. If the game is smooth, we have
>
> $$
>     \sum_{k=1}^T F^k=\Omega(T F^* - \sum_i R_i).
> $$
> Therefore, minimizing the summation of regret guarantees the dynamics converges to the optimal social welfare. A formal description can be found in Proposition 2 [2]. We have added a discussion in the revision to make it clear.
>
>  [1] Intrinsic robustness of the price of anarchy.
>
>  [2] Fast Convergence of Regularized Learning in Games
>
> > Obtaining convergence to a weak CCE via a pseudo-regret has been typically described in the literature? Is there any novelty in this point? I assume no, the ingenuity starts with the variance reduction methods, correct?
>
> Your understanding is correct. Our contribution is the variance-reduced algorithm.
>
> > The authors proposed LVE instead of classical IWE with exploration term. Can the authors explain the details of their estimator. It would help if they provide some extended answer. how they derive this estimator. Is it something standard in the literature?
>
> Thanks for your comments. As you commented, the LVE resolves the curse of dimensionality of IWE.
> In general, for two distributions $\sigma$ and $w$ over $M$ bins, it is impossible to develop low-variance estimator with $o(M)$ running time. In games, $M=$Cost or $|A|^{N-1}$ in the worst case. However, thanks to the structure of games, we are luckily to construct a low-variance ditribution which is a weighted mixing of $N-1$ distributions and dimensions are independent in each of these sub-distributions. Therefore, it admits fast sampling with running time $O((N-1)A)$. As far as we can tell, this is novel in the literature of both game theory and variance reduction.
>
> In addition, we have proved the variance and time complexity of LVE are near-optimal and optimal, respectively. Please see our response to reviewer ZKef and the revision for more details.
>
> > There is another point which was not clear, the regret statements are correct if all the players employ the same algorithm? Do we have the adversarial 1/\sqrt{T} case, if one of the players plays adversarially/stochastically?
>
> Thanks for your suggestion on discussing the adversarial and stochastic regret. We have added a subsection 6.3 in the revision to provide a $O(\sqrt{T})$ adversarial regret. As for the case of randomly selected strategy, regret with $O(\log T)$ order is out of the reach of this paper.

---

### Decision · Program_Chairs · 2023-01-20

**Decision:**

Accept: poster

**Justification For Why Not Higher Score:**

The authors did not justify why no-regret algorithms are relevant for the *computation* of CCE: even though no-regret learning is the norm from an online learning perspective, it is not clear why this would be relevant from a computational standpoint when the game is fully known to the computational device(s).

**Justification For Why Not Lower Score:**

The combination of variance reduction with mixed-strategy sampling is an idea that hasn't been sufficiently explored in the literature, and the paper provides an interesting study.

**Metareview: Summary, Strengths And Weaknesses:**

This paper studies the convergence of the optimistic mirror descent (OMD) method to coarse correlated equilibrium in general finite games.

The main bottleneck targeted by the authors is that the calculation of the players' mixed payoff vectors can be very costly (exponential in the number of players), so they instead utilize only the sampled payoff vectors (i.e., those consisting of the payoffs resulting from the players' actual actions). This device reduces the per-iteration complexity of OMD dramatically but, on the negative side, it also introduces randomness, which can reduce the overall oracle complexity of the algorithm. The authors mitigate this via a suitable variance reduction mechanism which achieves a CCE convergence rate close to the deterministic case (but with a lighter per-iteration cost). [The authors also examine convergence to weak-CCE, but this result was deemed to be less significant in scope]

The committee felt unanimously that the paper cleared the acceptance bar, so an accept recommendation was reached early on.

**Note From Pc:**

if the above contains the word "oral" or "spotlight" please see: "oral" presentation means -> notable-top-5% and "spotlight" means -> notable-top-25%. As stated in our emails, we are disassociating presentation type from AC recommendations

**Summary Of Ac-Reviewer Meeting:**

This was not a borderline paper.